# Identification and Analysis of the Key Factors That Influence Power Purchase Agreements on the Road to Sustainable Energy Development

Adrian Tantau [1,*], Elena Niculescu [2], Laurentiu Fratila [3], Costel Stanciu [4] and Cristina Alpopi [5]

1 UNESCO Department of Business Administration, Bucharest University of Economic Studies, 010371 Bucharest, Romania
2 Doctoral School Business Administration, Bucharest University of Economic Studies, 010371 Bucharest, Romania; bistriteanuelena09@stud.ase.ro
3 Department of Informatics, Bucharest University of Economic Studies, 010371 Bucharest, Romania; laurentiu.fratila@cig.ase.ro
4 Department of Economic Studies, University Nicolae Titulescu–Bucharest, 011141 Bucharest, Romania; costelstanciu@univnt.ro
5 Department of Public Administration, Bucharest University of Economic Studies, 010371 Bucharest, Romania; cristina.alpopi@ase.ro
* Correspondence: adrian.tantau@fabiz.ase.ro

**Abstract:** The analysis of the key factors which prevent or contribute to the promotion and development of power purchase agreements (PPAs) in order to meet sustainability goals represents an important issue which is worth being investigated, especially given the situation which is entered into considering the bilateral relations inside this contract, which is meaningful to achieving new climate targets, developing and improving the use of green energy, and promoting a sustainable green economy. The main goal of this research is to identify and analyse the main factors which either help or prevent the promotion and development of a PPA in order to meet sustainability goals by promoting a green economy. For this research, a survey based on comparative analysis, questionnaires and interviews with energy experts, Spearman's correlation matrix, and IBM SPSS Statistics for Windows are used. The results suggest that although there is a positive interest in and an open attitude towards PPA on both sides (sellers and buyers), there is still room for improvement; the lack of knowledge or insufficient measures taken so far are two of the reasons why, until now, in Romania, no PPA market has been developed, and the level of knowledge regarding this type of mechanism is quite low.

**Keywords:** power purchase agreement (PPA); key factors; sustainability goals; renewable energy; green energy; zero carbon economy; climate neutrality; Romania

## 1. Introduction

The problem of global warming has turned the attention towards renewable energy [1]. So, climate change and its implications are some of the most important issues faced by humanity [2], and from what many researchers say, the main cause of this is the burning of fossil fuels, which releases greenhouse gases (GHGs). Here, we mention that five industries (cement and concrete, iron and steel, oil and gas, chemicals, and coal mining) together are responsible for 80% of industrial emissions [3]. At the same time, taking into consideration the Paris Agreement, more world countries, including those within the European Union, are aiming to become carbon neutral by 2050 [4,5] to keep the level of global warming below 1.5 °C [6]. More actions aimed at reducing and even stopping greenhouse gas emissions derive from The Doha Amendment, adopted in Doha on 8 December 2012, the Kyoto Protocol, the United Nations Framework Convention on Climate Change, as well as the UN's climate change conference in Glasgow in 2021. Considering that between 2002

and 2030 the global energy demand will increase by approximately 60%, with an average increase of 1.7% annually, there will be an increase in greenhouse gas emissions [7]. It is thought that oil reserves could be exhausted by 2040, natural gas by 2060, and coal by 2300 [8]. In this sense, a safe and efficient way to be able to stop and defeat global pollution is to use as many renewable energy sources as possible [9], composed of sustainable and clean energy that comes from nature [10]. Of the same opinion is Moslem, S., 2024, [11] who considers that cities must have a sustainable and environmentally friendly answer in order to prevent this anticipated crisis. According to the International Energy Agency's experts, this could be a beneficial period for the development and use of renewable energy sources (RESs). It is expected that in the next 5 years, many renewable energy resources will be used, as were used in the last 20 years, thanks to the countries that are becoming leaders in this field [12]. In recent years, power purchase agreements (PPAs) have become better and better, receiving more and more importance; thus, at the end of 2014, the total number of farms that signed a PPA reached 363, and the total capacity has become 32,641 MW [13]. In addition to being an efficient way for energy producers from renewable energy sources to attract financing for the development of green energy infrastructure, PPAs are also a suitable way of encouraging the reallocation of resources, as well as a good instrument to achieve sustainability goals by promoting a green economy. Thus, PPAs represent an innovative and sustainable mechanism which plays an important role in environmental protection by encouraging an ecologically friendly responsibility and resulting in the achievement of climate sustainability by providing appropriate financial aid. The long-term stability of revenue streams helps producers improve solvency and acquire capital for their investments, as well as large consumers meet their sustainability objectives [14]. The analysis starts with PPA's definition. A power purchase agreement (PPA) is a performance-based contract for the purchase and sale of energy between an energy buyer and an energy provider, through which the energy buyer establishes the total amount of energy they want and the price agreed in the contract they will pay [15]. According to Vimpari, [16], in 2019, already 20 GWs from renewable energy source investments were developed based on PPAs signed between private companies. Another aspect to be mentioned is that one of the reasons why the PPA market in Romana is not well developed is because for almost 10 years, these contracts have been banned at the national level. Their reintroduction in Law no. 123/2012 regarding electricity and natural gas was possible because of the issuance and entry into force of EU Regulation no. 943/2019, which established the obligation of the Member States to regulate these contracts at the national level. In practice, the intervention of the European Union has been defining and boosted and promoted such contracts at the level of Member States. Thus, through GEO no. 143/2021, the Romanian government has created a legal framework for PPAs, as well as established more favourable conditions for net metering solar generation. Given that, at the national level, PPAs are not very well developed and promoted, with three such contracts signed so far, the decision was taken at the private level to organise conferences specifically dedicated to bringing these PPAs to the general public. Now in its second edition, the Romanian Green PPA is dedicated to using renewable energy and aims to facilitate meetings between producers, developers, renewable energy investors, consumers, suppliers, and energy traders. The purpose of this article is to identify and analyse the main factors which either help or prevent, in one way or another, the promotion and development of a PPA by promoting a green economy and to find appropriate measures that can be applied or improved. This article is structured as follows. The introduction presents the power purchase agreement concept. In Section 2, PPA is defined, with an analysis of the main scientific sources. Within Section 3, the research methodology is presented, which consists of a comparative analysis, a workshop with a panel of experts, a survey based on a questionnaire and Spearman's correlation matrix, the results of which are calculated using IBM SPSS Statistics for Windows and simple linear regression. This work's main purpose is to analyse and better understand the main factors which help or prevent the promotion and development of PPAs on the road to climate neutrality and meeting the related sustainability goals.

We formulated questionnaires mainly for the purpose of uncovering the elements that may affect PPAs in the transition to a zero-carbon economy, while a workshop made up of several energy experts was organised for the purpose of formulating the first hypothesis regarding the key factors that have an influence on the development or not of power purchase agreements as they appear to be from the questionnaire. Our hypotheses are as follows:

**Hypothesis 1.** *Experience is directly correlated with level of knowledge (considered independent variable) and influences the attitude towards the implementation of PPAs.*

**Hypothesis 2.** *The development of PPA is dependent on the level of how much the state is involved.*

**Hypothesis 3.** *The experience is independent of the party (seller/buyer). There are companies, mostly multinational, which have a large amount of experience from the buyer's point of view, which share with the Romanian branch the contracts that are already implemented in other countries.*

**Hypothesis 4.** *There is a direct correlation between energy consumed monthly and the benefits of signing a PPA. Therefore, signing a PPA is dependent on the amount of energy consumed monthly, so the attitude towards PPAs is dependent on monthly consumption.*

Section 4 presents the main research results and, last but not least, the final section shows the main conclusions. The bibliographical references complete this article.

## 2. Power Purchase Agreement in the Scientific Literature

The desire of European leaders is to develop renewable energy sources in order to create a greener planet and a sustainable green economy. According to Wang [17], investing in renewable energy sources will reduce their carbon footprint and avoid large climate problems. At the same time, due to Belgacem's [18] opinion, there is a need for more regulations in the renewable energy field because there are many financial constraints for developing countries. In this regard, the encouragement of using renewable energy falls on governments, who are best placed to ensure that there are regulations aimed at ensuring that the transition to green energy is performed in a responsible and sustainable way.

Many companies willing to achieve climate neutrality are implementing green financial methods, designed to use much more green energy and reduce carbon emissions. These financial and green purposes could be easily achieved through power purchase agreements (PPAs), which are contracts that ensure the supply of electricity for distribution companies from a generation capacity. In order to be concluded, these contracts require the establishment of a fixed rate for a certain amount of energy to be purchased by a buyer.

In addition to the abovementioned aspects, it must be added that according to Liobikiene and Miceikiene, [19], The European Commission, through the Green Deal, focused on achieving a carbon-neutral or climate-neutral EU economy by 2050, aims to make environmental policies involving climate change as well as a circular economy, among them also including PPAs.

The main characteristic of a renewable PPA is that a certain amount of energy can be sold from a renewable project, which means that signing a long-term PPA is very important for any renewable project because it secures a long-term stream through the sale of energy at a sure price [20], having the benefits of using green or renewable energy, which means taking a step forward towards the transition to green energy by reducing CO2 emissions and achieving, at the same time, climate neutrality. Approximately 45% (20.9 GW) of corporate PPAs announced in 2023 were concluded in the Americas region, followed by Europe with 33% (15.4 GW). Between 2022 and 2023, Europe saw a 74% increase in corporate PPA volumes to 15.4 GW, by far the largest increase of all the regions [21].

At the same time, it must be highlighted that in Europe, PPAs have been signed more in Scandinavia, Great Britain, Spain, and the Netherlands, and recently in Italy. The most used renewable energy resource is wind energy, with the exception of Spain, where the most commonly used energy is that from solar sources [22].

Although many studies have been performed on the assessment of climate policy and emission reductions, according to Kumar [23], these studies have not assessed these types of contracts; the payment arrangements in PPAs establishing the costs of power transmission have to be as clear as possible. According to Tranberg [24], there is a negative dependence between wind power production and electricity spot price, this being a relevant factor which can be taken into consideration when we talk about the risk management of long-term PPAs. In their article, Taghizadeh-Hesary [25] affirmed that PPAs are the most appropriate way of attracting green investment, and they draw attention on the private system that should be more involved in taking part in reducing pollution. In this situation, PPAs are considered the most effective tools that can help develop green energy and the most efficient mechanism that can improve the relation between a clean energy buyer and a clean energy supplier. At the same time, according to Xiang [26], under the effects of PPAs, energy producers are much more motivated to take effective measures in developing the amount of energy produced.

The mechanism of PPA means, first of all, for the buyers that there is no need to build and maintain energy-producing equipment, making it more convenient to buy this energy directly from sellers through a PPA.

Secondly, the PPA brings benefits also for producers [27], in the sense that they will have a constant flow of energy that will be paid for according to a price mutually agreed upon with the buyer. It should be mentioned that the minimum and maximum of the delivered energy are set by the buyer, and once the maximum amount is reached, they can choose if they want to buy an additional amount of energy at a lower price. Otherwise, when the established amount of energy is not reached, the producer will have to buy the remaining quantity from the market and sell it to the buyer.

At the same time, PPAs bring another advantage for renewable energy producers due to their price security for a forthcoming unsecure energy amount, while the buyers sign PPAs due to the durability they offer. However, for buyers, the insecurity of the energy quantity produced by renewable energy producers can bring both economic and technological difficulties [28,29]. Another economic benefit worth being mentioned is that of reducing maintenance costs, improving delivery power, offering long-term price security, offering opportunities to finance investments in power generation capacities, and, last but not least, reducing the risks associated with electricity sales and purchases. When a specific physical supply of electricity with certain regional characteristics and guarantees of origin occurs, this is an opportunity that can make a brand more sustainable and greener [30]. Another important economic advantage is the price, which can be fixed, established under the form of contracts, meaning that it can be different or even variable.

In conclusion, among the economic benefits that help PPAs develop the sustainability of the business environment, the following should be included: they help to develop investments in renewable energy projects; they secure long-term energy prices against the volatility of market energy prices; they ensure long-term cash flows; entering into PPAs is in itself an environmentally sustainable development strategy; and they help reduce carbon footprint, moving towards achieving a zero-emissions economy through the use of renewable energy.

Important to be remembered is the fact that PPAs have the possibility to reduce the risks which may occur during the project for both parties (the off-taker and the producer) and to increase the growth of renewables [31]. As it is mentioned in IRENA's 2018 report, PPAs are also very important to regulate the most considerable aspect from the contract, meaning the price, as well as the responsibilities between parties and the associated risks.

Furthermore, this contract is an instrument that fights against the electricity market's price volatility in developing economies, because different aspects are taken into consid-

eration during the negotiations. Here, we can remember the budget of the off-taker, the capacity of green energy that can be produced, the type of renewable resources that can be used, government legal measures, and the technologies requested to generate power. These are some of the advantages as to why the development of green policies can be seen and why the interest in signing such a contract is growing. It happens also that a purchased energy limit is obtained, meaning that the off-taker has the possibility of not paying any more than the required contractual price, and excess energy could be bought at a reduced price or not at all. This is the moment when the seller has the possibility (according to some PPAs) to sell the energy that is in excess into the spot market, where they can obtain lower or higher prices [32]. According to Gabrielli [33], participants in the market, such as energy utilities, traders, and electricity market operators, prefer the contractual forms that offer the possibility of several income flows, but also to offer control over the generation and storage of technologies. On the other hand, the corporate buyers prefer PPAs due to their possibility of avoiding market volatility to reduce energy costs and pollution and to gain sustainability. For them, simple contract structures are favourable, namely the ones where they do not have full control of the generation or storage of assets, such as the following:

1.  Tolling agreement—This structure offers the buyer the possibility of controlling the storage and to function within multiple markets, for example ancillary services, intraday arbitrage, and day-ahead market arbitrage. At the same time, the producer receives the energy price in order to pay for the operational costs and capacity payments to cover their fixed costs [34]. In Germany, this structure is used more by energy traders and utilities, which have more expertise in energy markets.
2.  Energy contracts—According to [34], through this structure, the buyer pays a fixed price for the energy produced by an RE plant coupled with energy storage technology. Through this type of contract, the buyer benefits from day-ahead market revenues, while the producer takes advantage of offering ancillary services and playing intraday arbitrage.
3.  RE store contracts—This type of contract is based on the difference between the highest and lowest price hours of the day-ahead auction [35]. For corporate buyers, this contract is appropriate due to the independency of the cash flows from the operation of the storage asset and established on day-ahead market prices. This type of contract notes the revenues of grid-charged storage, but if it is desired to retain in the contract the various round-trip efficiencies of the storage technology or if it is desired to note the revenues from wind or solar storage, this is not possible, so this contract cannot be adapted for these purposes.

## 3. Research Methodology

PPAs are designed to protect energy producers, suppliers, and consumers, playing a beneficial role in both long-term price security and the development of new renewable energy capacity.

However, precisely because the contractual period of PPAs is longer, entering into such relationships involves taking risks, but also understanding some of the factors that may influence the validity of the contract. In this respect, it is important that factors that may take the form of drivers (if they help promote PPAs) or barriers (if they hinder this promotion) are known and analysed in as much detail as possible, this being the main motivation of this research.

As far as the contribution of this paper is concerned, we consider it to be extremely important in terms of understanding how PPAs operate in relation to the external factors that positively or negatively influence entry into such contractual relationships.

By better understanding these drivers/barriers, stakeholders can have a better overview of what they should consider or expect when making the decision to enter into a PPA, as the long period of time involved in these contracts forces parties to make a more rigorous and thorough analysis, which makes the knowledge of factors that help or hinder the promotion of PPAs all the more important.

The research methodology used in this paper described in Figure 1, consists of a comparative analysis, a workshop with a panel of experts, and a survey based on a questionnaire and Spearman's correlation matrix, the results of which are calculated using IBM SPSS Statistics for Windows (Version 29.0.0.0, Software vendor: https://www.ibm.com/products/spss-statistics, accessed on 7 March 2024) and simple linear regression. This work has the main purpose of analysing and better understanding the main factors which help or prevent the promotion and development of PPAs on the road to climate neutrality and meeting the related sustainability goals.

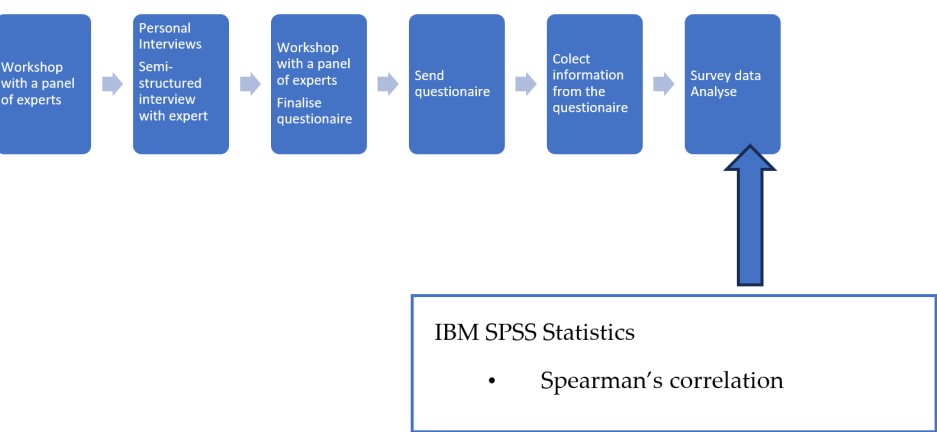

**Figure 1.** The steps of the methodology (Source: authors' elaboration).

The research method is that of comparative analysis with the objective, as we previously outlined, of determining the key factors that help or prevent the development of PPAs and the relationship between them. A second method of research is the organisation of a workshop made up of several energy experts which had the purpose of formulating the first hypothesis regarding the key factors that have an influence upon the development or not of PPAs as they appear to be from the questionnaire.

The participants included the following: an energy trader, responsible for purchasing a portfolio of the top 10 universal customer suppliers in Romania, a developer of renewable projects, a renewable electricity producer, and an average industrial consumer (category C3). These experts were randomly selected from the list of respondents from the questionnaire, without knowing their answers, their role being to expose their opinion on PPAs' key factors based on the questionnaire results.

All hypothesis presented in the Introduction will be analysed from two points of view: the seller's and buyer's perspective.

We formulated questionnaires mainly to discover the elements that may affect the PPAs in the transition to a zero-carbon economy, while their results are processed with IBM SPSS Statistics for Windows ((Version 29.0.0.0, Software vendor: https://www.ibm.com/products/spss-statistics, accessed on 7 March 2024). The data obtained through the formulation of the questionnaire revealed that the main factors which influence or prevent the development of PPAs are as follows (Figure 2):

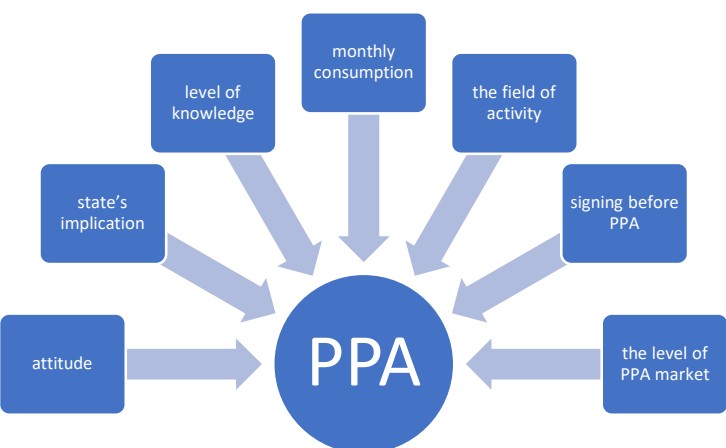

**Figure 2.** Key factors that influence the development of PPA (Source: authors' elaboration).

It has to be mentioned that the survey involved 232 respondents, and we decided to determine the factors which contribute to or prevent the development of PPAs on the road to climate neutrality and meeting the related sustainability goals. Regarding the respondents, we emphasise that they fall into different categories, representing different fields of activity, such as economic operators and industrial and small consumers. The confidentiality of the respondents is respected throughout the study. Personal data were not the subject of this study, and the questions provided are not subjective in nature. Furthermore, aspects such as attitude, state's implication, level of knowledge, the level of PPA market, if the respondents have ever signed a PPA, the field of the company, and monthly consumption were taken into account when analysing the results.

The profiles of the consumers who answered the questionnaires are presented in Figure 3.

|  |  | Frequency | Percent |
|---|---|---|---|
| Parties | Buyer | 167 | 72.0 |
|  | Seller | 60 | 25.9 |
|  | Other | 5 | 2.2 |
| Level of knowledge | Very low | 75 | 32.3 |
|  | Low | 59 | 25.4 |
|  | Medium | 74 | 31.9 |
|  | High | 19 | 8.2 |
|  | Very high | 5 | 2.2 |
| Signing before PPA | Yes | 55 | 23.7 |
|  | No | 177 | 76.3 |
| Monthly energy consumption | 0 – 50 MWh | 98 | 42.2 |
|  | 50 – 100 MWh | 59 | 25.4 |
|  | 100 – 500 MWh | 64 | 27.6 |
|  | >500 MWh | 11 | 4.7 |

**Figure 3.** Respondent profile. Source: authors' elaboration on sample data.

In order to obtain the research results, the data were processed with IBM SPSS Statistics for Windows (Version 29.0.0.0, Software vendor: https://www.ibm.com/products/spss-statistics, accessed on 7 March 2024). The statistical analysis performed as a result of the processing of the questionnaire was based on the following stages: collection, processing, analysis, and interpretation of the survey results. Regarding the horizontal analysis, we show that the answers to each question have different units of measure (ordinal, nominal, scale), so the horizontal analysis is not strong. The vertical analysis aimed to achieve statistical correlations in order to explore the presence and intensity between the variables

included in the model. We wanted to discover correlations between different factors which influence risks and whether these factors were strongly correlated to other factors.

Spearman's rank correlation was used. The main variables used for the correlation analysis are as follows: attitude, level of knowledge, experience, state's implication, the level of the PPA market, if the respondents have ever signed a PPA, the field of the company, and monthly energy consumption.

## 4. Results and Discussion

As we mentioned in the previous chapter, the analysis was carried out based on the questionnaire answered by 232 respondents, their profile being described according to the table below:

The buyer represents the person in the contract that purchases the energy from the seller, who in this contract is represented by the renewable energy producer.

Frequency represents the number of each answer taken into consideration for the descriptive profile.

Percent is the share of each answer from the total number of respondents.

Signing before PPA describes a previous signing of such an agreement for the respondent.

From the analysis in Figure 3, it can be seen that most of the respondents have the quality of a buyer within the PPA with a percentage of 72%, while only 60 respondents have the quality of a seller, with a total percentage of 25.9%. In terms of level of knowledge, it can be seen that the majority of respondents are people who have very low (32.3%), low (25.4%), and medium (31.9%) levels of knowledge, while the difference up to 100% is divided as follows: high level of knowledge, 8.2%, and very high level, only 2.2%. Regarding the aspects related to the situation of signing before PPAs, it can be observed that 55 (23.7%) respondents have signed PPAs before, while the rest, 177 (76.3%), have not entered into such contractual relations. Correlating the aspects indicated in points 2 and 3 in the table presented above, we can see that even if 10.5% of the respondents have so far signed a PPA, it can still be observed that almost half of the respondents (42.3%) have, on average, a high or very high level of knowledge, which shows a fear of entering into a PPA; on the one hand, it also shown an urgency on the part of the state to take the necessary measures to encourage entry into such commercial relationships. Last but not least, the analysis of the respondents' profile must also be correlated with monthly energy consumption, where it can be seen that compared to the companies represented by the respondents, it appears that they are small and medium industrial consumers, with a monthly consumption of less than 500 MWh, adding up to a total percentage of 95.3%.

The rank and effect ratio are based on the highest percentage answer of each individual factor compared with the rest. For example, if for one factor we have an answer that collected 70% of the respondents and for another factor the highest percentage is 30%, from the global impact on PPAs, the first factor has 70% and the second 30%. Results are presented in Figure 4 and methodology is presented in Figure 5.

The low level of development of these contracts can be seen from the analysis of the answers given by the respondents, with the key factors analysed in this article having an impact on signing PPAs, as shown in Table 1.

**Table 1.** Impact of each key factor on signing PPA.

| Number | Key Factors | Percentage |
|--------|-------------|------------|
| 1. | Signing before PPAs | 19% |
| 2. | Level of PPA market | 19% |
| 3. | Attitude | 16% |
| 4. | State's implication | 14% |
| 5. | Field of activity | 13% |
| 6. | Monthly consumption | 11% |
| 7. | Level of knowledge | 8% |

Source: authors' elaboration on sample data.

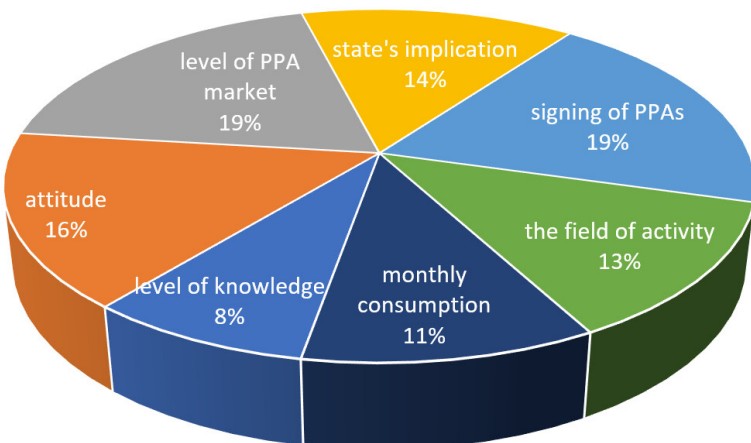

**Figure 4.** Key factors—their weight in signing a PPA. Source: authors' elaboration on sample data.

| Factor | Highest % | % of global impact of the Total factors (Tf) |
|---|---|---|
| Signing before PPAs | 76.3 | 19% |
| Level of PPA market | 75.4 | 19% |
| Attitude | 62.9 | 16% |
| State's implication | 55.6 | 14% |
| Field of activity | 50 | 13% |
| Monthly consumption | 42.2 | 11% |
| Level of knowledge | 32.3 | 8% |
| Global Impact of the Total factors (Tf) | 394.7 | 100% |

**Figure 5.** The method of calculating the impact that the key factors have on the signing of PPAs. Source: authors' elaboration on sample data.

From the analysis of the percentages shown in Figure 5, we can observe that the first place is equally occupied by the previous signing by the respondents of PPAs, respectively, by the level of the PPAs market, which leads us to the obvious conclusion that both the signing of these contracts and their market level significantly contribute to the development and promotion of PPAs. In second place, with a percentage of 16% is the attitude of the respondents towards PPAs. Analysing this percentage by referring to the results obtained and described in the previous chapter, we can see that the effect of this factor on the signing of PPAs is a beneficial one, with the attitude of the respondents being favourable (65% of the buyers are in favour of such contracts) and open towards these contracts. Third place is occupied by the factor representing the involvement of the state, with the percentage allocated to it being, as can be seen, 14%. Thus, it is clear that the level of state involvement plays an important role in the promotion and development of PPAs, being on the podium in the analysis of the impact that key factors have on the signing of these contracts. Regarding the other factors, even if they have an impact on signing a PPA, their effect is not a defining one, taking into consideration that 54% of the choice of signing a PPA is taken based on the first three factors.

Additionally, it should be mentioned that the results of the questionnaire were analysed from two perspectives: the perspective of the buyer and the perspective of the seller, and they are presented in the graphic illustrations in Figures 6–9.

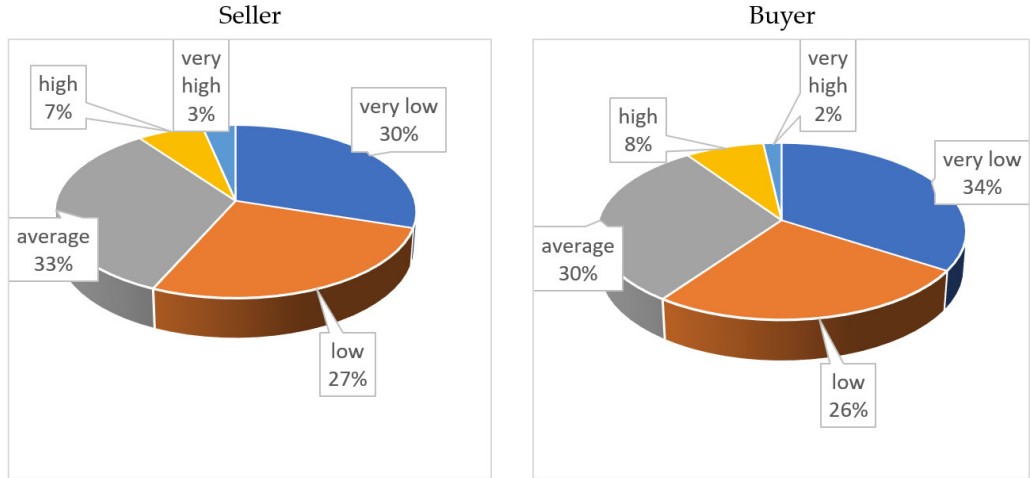

**Figure 6.** The level of knowledge. Source: authors' elaboration on sample data.

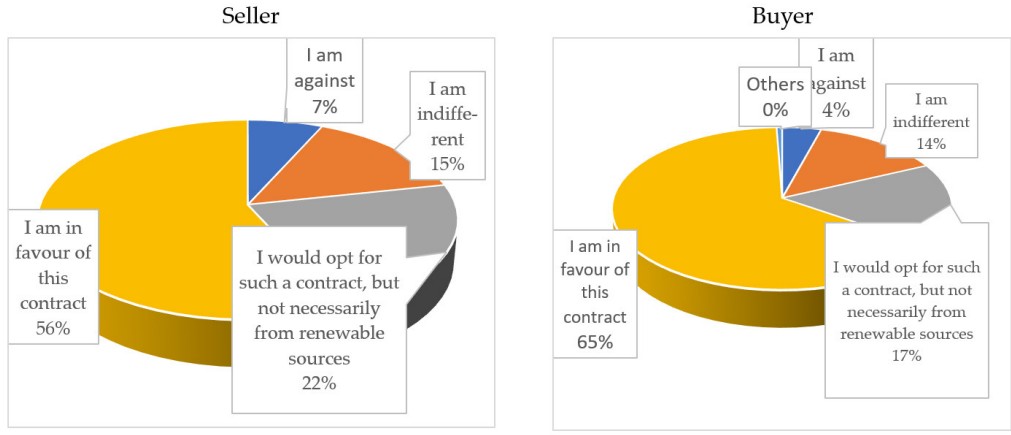

**Figure 7.** Attitude. Source: authors' elaboration on sample data.

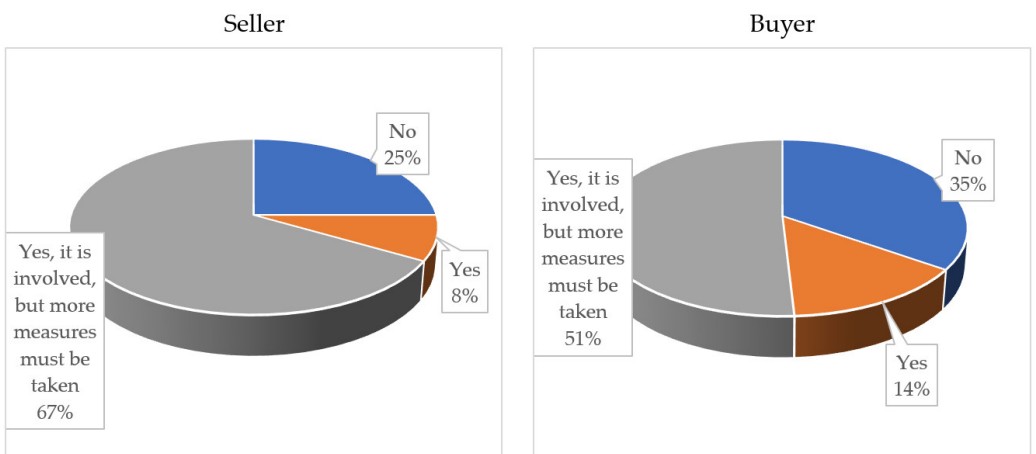

**Figure 8.** State's implication. Source: authors' elaboration on sample data.

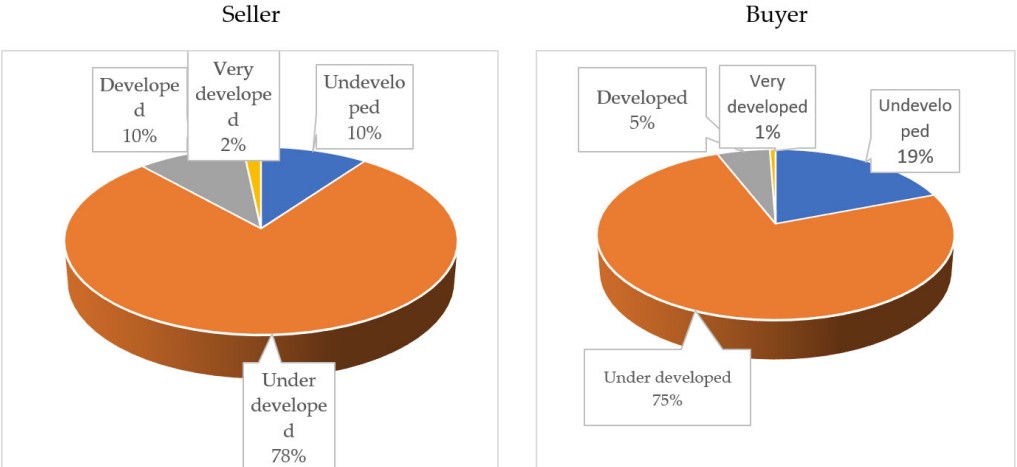

**Figure 9.** The level of PPA market in Romania. Source: authors' elaboration on sample data.

From the analysis of the diagram above, we can see that with regard to the level of knowledge between the two parties (buyer and seller), the values are quite similar; we are thus talking about a percentage of, in total, only 10% (in the case of the seller), who own a high level of knowledge (7%) and very high level (3%), respectively. In the case of the buyer, this is divided slightly differently 8%—high level of knowledge and 2%—very high level of knowledge. On the opposite pole, we are talking about a lack of knowledge about PPAs, where it is found that more than half of the interviewees have a low or very low level of knowledge: in the case of the seller, we have a total percentage of 57% (30% very low, 27% low), so that in the case of the buyer, there is a percentage of about 60% divided in this way: 26% have a low level of knowledge and 34% have a very low level of knowledge, respectively. All these results are somehow concerning, because a very low level of knowledge can be proven both in terms of the buyer and in terms of the seller, even though, according to Lei [13], PPAs have gone from strength to strength, because at the end of 2014, the total number of farms that signed PPAs reached 363, and the total capacity reached 32,641 MW. Analysing all the above mentioned aspects, we can observe that in Romania, there is a very low level of knowledge regarding the existence of PPAs, regardless of whether we are talking about buyers or sellers. Thus, we ask ourselves how this field can be developed when very few people who work within it know what PPA means or at least have heard of it, even less the persons who do not have information about and who are not involved in the area of energy? On the other hand, the low and very low level of knowledge, where more than half of the respondents do not have information and do not know about the existence of the PPA, must be analysed by comparison with the involvement of the state, the only one able to make all efforts to find out the level at which PPAs are located in Romania, respectively, to find out their degree of development, being, at the same time, the only one that has the opportunity to take all the necessary measures to help develop and subsequently implement this type of contract.

Regarding attitude, we can see that the results are somewhat gratifying; more than half of the respondents are in favour of concluding such contracts. Thus, we are talking about a percentage of 56% in the case of sellers and a percentage of 65% in the case of buyers, respectively, the latter percentage proving, without a doubt, that these respondents have a much more open attitude towards the signing of PPAs for the purchase of energy from renewable energy sources. In second place are the respondents who consider that they would opt for such a contract, but the purchased energy does not necessarily have to be from renewable energy sources, which means that they did not accurately understand the true role and true meaning of a PPA as an instrument that enables the meeting of related sustainability goals. Thus, although both sides have a positive attitude regarding the conclusion of PPAs, we can see that this attitude is based on a misunderstanding of the true role of this mechanism, which is a useful tool for the promotion and development of

the use of energy from renewable sources and not just a simple contract through which electricity is purchased.

Regarding the abovementioned aspects, the experts found the distribution interesting, showing on the one hand the importance of PPAs, and on the other hand the urgency of taking the needed actions towards the development of such contracts. But now a question arises: what made the respondents have such a positive attitude towards PPAs, as we can see from the answers revealed in Figure 2 that the level of knowledge among the respondents is low, with more than half of them having less knowledge of PPAs? But regardless of the reasons that were the basis of the answers, the situation is a gratifying one, and the fact that such a large number of the interviewed people consider these contracts to be beneficial can only be encouraging that in the near future, they can even be implemented.

According to the respondents, the state's level of involvement is visible, but it is not sufficient, and it is clearly necessary to take more measures to promote and develop this mechanism. Comparatively, by analysing the results both in terms of buyers and sellers, we find that the percentage in terms of the second category is much higher, with 67% of them considering that although the involvement of the state is positive, it is clear that there is a need to create more facilities to support renewable energy producers. According to the opinion of the experts, these results should be correlated with the results given to the attitude question (Figure 7), from where it can be seen that although the respondents are in favour of these contracts and they have a positive attitude towards them, the involvement of the state is only at a beginner level. As far as we can see, even if 76% of the respondents from the seller category and 51% from the buyer category, respectively, consider that the state is involved, this involvement is not sufficient, needing much more support and much more allocation of time and resources. Thus, by corroborating the result mentioned above with the low and very low level of knowledge of the respondents, we can clearly state that the measures taken at the state level are insufficient; the increased involvement of the legislator is not only necessary but urgent. As a conclusion from these results, we can say that the state, although it took certain measures, besides the fact that they were not sufficient (and here we refer to both categories), have not taken enough measures. It can be seen that our conclusion is also in line with the opinion of Belgacem [13], who considers that more regulation is needed in the field of renewable energy, as there are many financial constraints for developing countries. Thus, governments need to encourage the use of renewable energy resources and ensure that there are regulations in place to guarantee that the transition to green energy is achieved in a responsible and sustainable manner.

The results show, in an almost overwhelming majority, that the level of development of PPAs in Romania is very low, with 78% of the respondents who have the status of sellers and 78% of the respondents who have the status of buyers, respectively, having this opinion. Moreover, if we add the percentage of those who consider the market to be undeveloped (10% in the case of sellers and 19% in the case of buyers, respectively), we can see that almost unanimously we are talking about a very low level of the PPA market in Romania. This situation draws attention to the urgent need to take necessary measures regarding the impetuous obligation of the state to proceed with the promotion and development of this mechanism on the territory of the country. Noting the very small percentages regarding the respondents' appreciation of the existence of a developed and highly developed market (12% in the case of sellers and 6% in the case of buyers, respectively) only reinforces the idea presented in the previous paragraph regarding the urgency of taking measures at the state level. At the same time, the adoption of new technologies, including the increase in productivity, will allow for an efficient increase [36]. In this moment, a short comparison between attitude and the level of the PPA market must be made: regarding attitude, even if we can see that this is a positive one, with more than half of the respondents from both the buyer's side (65%) and the seller's side (56%) being open to signing PPAs, this situation must be analysed in relation to the development level of the PPAs market, where, as we can see, the situation is different, with the results being weaker. In this sense, it can be observed that 75% of the respondents who have the quality of buyers and 78% of the

respondents who have the quality of sellers consider that the market is an undeveloped market, while 10% of the sellers and 19% of the buyers appreciate that the level of the market is an underdeveloped one, which means there is a total percentage of 88% (seller) and 96% (buyer) who are of the opinion that the PPA market in Romania is a poorly developed one. A fast conclusion that can be drawn from this analysis demonstrates the urgency and necessity of the measures to be taken. Despite the information presented above, in this present moment, we can see an attempt to recover the energy market in Romania, which appears to be a competitive and dynamic one, and although it does not appear to be a mature market like the other western European markets, it is evolving quickly and gives a sign of increased competitiveness [37]. As far as it is known, PPAs are an effective and viable tool that, according to specialised authors, offer long-term sustainability and safety. According to [29], for buyers, the insecurity of the amount of energy produced by renewable energy producers can bring both economic and technological difficulties. Also, Taghizadeh–Hesary [25] proved that PPAs represent a sustainable and green tool in that they offer long-term security and an investment financing opportunity. In accordance with the opinion mentioned above, there are also the assessments of Mendicino [30], according to who PPAs have the possibility to increase the growth of renewables. This is also the reason why the corporate PPA market (in which large, industrial, electricity-consuming companies increasingly wish to conclude contracts with energy producers from renewable energy sources) is increasingly developing at the European level.

Regarding the buyer, Spearman's rank correlations, as illustrated in Table 5, show that the higher the level of knowledge, the greater the respondents' experience in the field of signing and implementing PPAs. The results revealed a statistically relatively strong relationship between the level of knowledge and the experience that respondents have regarding the PPAs field (correlation coefficient: 0.458). This result proves the importance of the level of knowledge being as high as possible; thus, the higher the level of knowledge regarding PPAs, the more the respondents are involved or the more information is brought to their attention regarding these type of contracts, thus increasing their experience, inevitably implying the development of this field. This emphasises that the first hypothesis is partially correct as long as we can see a correlation between the experience and level of knowledge, but the second part of the first hypothesis proved to be false, as shown below, in Tables 3 and 4:

**Table 2.** Buyer's correlation matrix (Spearman's rho correlation).

| | | Level of Knowledge | Attitude | State's Implication | PPA Market | Experience | Have You Ever Signed a PPA? | The Company Works in the Energy Field | The Monthly Energy Consumption |
|---|---|---|---|---|---|---|---|---|---|
| Level of knowledge | Correlation Coefficient | 1.000 | 0.235 ** | 0.068 | 0.245 ** | 0.458 ** | 0.009 | −0.134 | 0.214 ** |
| | Sig. (2-tailed) | | 0.002 | 0.386 | 0.001 | 0.000 | 0.909 | 0.085 | 0.006 |
| | N | 167 | 167 | 167 | 167 | 167 | 167 | 167 | 167 |
| Attitude | Correlation Coefficient | 0.235 ** | 1.000 | 0.093 | 0.129 | 0.226 ** | 0.227 ** | −0.036 | 0.072 |
| | Sig. (2-tailed) | 0.002 | | 0.231 | 0.097 | 0.003 | 0.003 | 0.641 | 0.355 |
| | N | 167 | 167 | 167 | 167 | 167 | 167 | 167 | 167 |
| State's implication | Correlation Coefficient | 0.068 | 0.093 | 1.000 | 0.094 | −0.025 | 0.046 | 0.006 | 0.091 |
| | Sig. (2-tailed) | 0.386 | 0.231 | | 0.225 | 0.745 | 0.551 | 0.939 | 0.240 |
| | N | 167 | 167 | 167 | 167 | 167 | 167 | 167 | 167 |

**Table 2.** *Cont.*

| | | Level of Knowledge | Attitude | State's Implication | PPA Market | Experience | Have You Ever Signed a PPA? | The Company Works in the Energy Field | The Monthly Energy Consumption |
|---|---|---|---|---|---|---|---|---|---|
| PPA market | Correlation Coefficient | 0.245 ** | 0.129 | 0.094 | 1.000 | 0.241 ** | −0.062 | −0.030 | 0.059 |
| | Sig. (2-tailed) | 0.001 | 0.097 | 0.225 | | 0.002 | 0.425 | 0.697 | 0.447 |
| | N | 167 | 167 | 167 | 167 | 167 | 167 | 167 | 167 |
| Experience | Correlation Coefficient | 0.458 ** | 0.226 ** | −0.025 | 0.241 ** | 1.000 | −0.124 | −0.565 ** | 0.009 |
| | Sig. (2-tailed) | 0.000 | 0.003 | 0.745 | 0.002 | | 0.111 | 0.000 | 0.907 |
| | N | 167 | 167 | 167 | 167 | 167 | 167 | 167 | 167 |
| Have you ever signed a PPA? | Correlation Coefficient | 0.009 | 0.227 ** | 0.046 | −0.062 | −0.124 | 1.000 | −0.009 | 0.021 |
| | Sig. (2-tailed) | 0.909 | 0.003 | 0.551 | 0.425 | 0.111 | | 0.903 | 0.784 |
| | N | 167 | 167 | 167 | 167 | 167 | 167 | 167 | 167 |
| The company works in the energy field | Correlation Coefficient | −0.134 | −0.036 | 0.006 | −0.030 | −0.565 ** | −0.009 | 1.000 | 0.008 |
| | Sig. (2-tailed) | 0.085 | 0.641 | 0.939 | 0.697 | 0.000 | 0.903 | | 0.921 |
| | N | 167 | 167 | 167 | 167 | 167 | 167 | 167 | 167 |
| The monthly energy consumption | Correlation Coefficient | 0.214 ** | 0.072 | 0.091 | 0.059 | 0.009 | 0.021 | 0.008 | 1.000 |
| | Sig. (2-tailed) | 0.006 | 0.355 | 0.240 | 0.447 | 0.907 | 0.784 | 0.921 | |
| | N | 167 | 167 | 167 | 167 | 167 | 167 | 167 | 167 |

Source: authors' elaboration on sample data (N—number of respondents, Sig (2-tailed) < 0.05)). ** Correlation is significant at the 0.01 level (2-tailed).

**Table 3.** ANOVA Table with Attitude regression for buyer.

| | | ANOVA [a] | | | | |
|---|---|---|---|---|---|---|
| Model | | Sum of Squares | df | Mean Square | F | Sig. |
| 1 | Regression | 9.863 | 2 | 4.931 | 6.671 | 0.002 [b] |
| | Residual | 121.227 | 164 | 0.739 | | |
| | Total | 131.090 | 166 | | | |
| 2 | Regression | 8.498 | 1 | 8.498 | 11.438 | <0.001 [c] |
| | Residual | 122.592 | 165 | 0.743 | | |
| | Total | 131.090 | 166 | | | |

[a] Dependent variable: attitude; [b] predictors: (constant), experience, level of knowledge; [c] predictors: (constant), level of knowledge.

At the same time, we can see a significant correlation between the field in which the company works and experience (0.565). This demonstrates that working in the energy field or having some responsibilities involving the energy field has a major impact on the experience of the respondents and therefore the possibility of entering or not entering into contractual relations within the PPA.

Regarding the involvement of the state, it is to be noticed that there is no clear correlation between it and the other analysed factors, which shows that hypothesis 2 is not confirmed.

**Table 4.** Coefficients Table with Attitude regression for buyer.

| | Model | Unstandardised Coefficients | | Standardised Coefficients | t | Sig. | 95.0% Confidence Interval for B | |
|---|---|---|---|---|---|---|---|---|
| | | **B** | **Std. Error** | **Beta** | | | **Lower Bound** | **Upper Bound** |
| 1 | (Constant) | 2.883 | 0.166 | | 17.386 | <0.001 | 2.555 | 3.210 |
| | Level of knowledge | 0.174 | 0.071 | 0.205 | 2.460 | 0.015 | 0.034 | 0.314 |
| | Experience | 0.087 | 0.064 | 0.113 | 1.359 | 0.176 | −0.039 | 0.212 |
| 2 | (Constant) | 2.967 | 0.154 | | 19.259 | <0.001 | 2.663 | 3.272 |
| | Level of knowledge | 0.216 | 0.064 | 0.255 | 3.382 | <0.001 | 0.090 | 0.342 |

[a] Dependent variable: attitude. Source: authors' elaboration on sample data.

The analysis of the correlations in IBM SPSS Statistics for Windows (Version 29.0.0.0, Software vendor: https://www.ibm.com/products/spss-statistics, accessed on 7 March 2024) through the Spearman method also revealed that there is no correlation between the attitude towards PPAs and level of knowledge (correlation coefficient 0.235) and between monthly energy consumption and attitude, where the correlation coefficient is 0.072, which leads us to the idea that hypothesis 4 is false.

Regarding the Spearman's rho correlations from the seller's point of view, when we analyse the results showed in the Table 5, we can definitely see that there is no correlation between the variables taken into consideration, with the correlation coefficient in all the cases being very low, a situation that does nothing but prove that all our hypotheses are not confirmed.

**Table 5.** Seller's correlation matrix (Spearman's rho correlation).

| | | Level of Knowledge | Attitude | State's Implication | PPA Market | Experience | Have You Ever Signed a PPA? | The Company Works in the Energy Field | The Monthly Energy Consumption |
|---|---|---|---|---|---|---|---|---|---|
| Level of knowledge | Correlation Coefficient | 1.000 | 0.348 ** | 0.168 | 0.201 | 0.305 * | −0.192 | −0.130 | 0.176 |
| | Sig. (2-tailed) | | 0.006 | 0.201 | 0.123 | 0.018 | 0.142 | 0.320 | 0.180 |
| | N | 60 | 60 | 60 | 60 | 60 | 60 | 60 | 60 |
| Attitude | Correlation Coefficient | 0.348 ** | 1.000 | 0.373 ** | 0.154 | 0.135 | 0.181 | −0.174 | −0.214 |
| | Sig. (2-tailed) | 0.006 | | 0.003 | 0.241 | 0.305 | 0.165 | 0.183 | 0.100 |
| | N | 60 | 60 | 60 | 60 | 60 | 60 | 60 | 60 |
| State's implication | Correlation Coefficient | 0.168 | 0.373 ** | 1.000 | 0.230 | −0.045 | 0.337 ** | −0.099 | 0.151 |
| | Sig. (2-tailed) | 0.201 | 0.003 | | 0.076 | 0.731 | 0.008 | 0.453 | 0.249 |
| | N | 60 | 60 | 60 | 60 | 60 | 60 | 60 | 60 |
| PPA market | Correlation Coefficient | 0.201 | 0.154 | 0.230 | 1.000 | 0.087 | 0.104 | −0.027 | 0.021 |
| | Sig. (2-tailed) | 0.123 | 0.241 | 0.076 | | 0.509 | 0.429 | 0.836 | 0.876 |
| | N | 60 | 60 | 60 | 60 | 60 | 60 | 60 | 60 |

**Table 5.** *Cont.*

| | | Level of Knowledge | Attitude | State's Implication | PPA Market | Experience | Have You Ever Signed a PPA? | The Company Works in the Energy Field | The Monthly Energy Consumption |
|---|---|---|---|---|---|---|---|---|---|
| Experience | Correlation Coefficient | 0.305 * | 0.135 | −0.045 | 0.087 | 1.000 | −0.282 * | −0.385 ** | 0.263 * |
| | Sig. (2-tailed) | 0.018 | 0.305 | 0.731 | 0.509 | | 0.029 | 0.002 | 0.043 |
| | N | 60 | 60 | 60 | 60 | 60 | 60 | 60 | 60 |
| Have you ever signed a PPA? | Correlation Coefficient | −0.192 | 0.181 | 0.337 ** | 0.104 | −0.282 * | 1.000 | −0.021 | −0.002 |
| | Sig. (2-tailed) | 0.142 | 0.165 | 0.008 | 0.429 | 0.029 | | 0.872 | 0.987 |
| | N | 60 | 60 | 60 | 60 | 60 | 60 | 60 | 60 |
| The company works in the energy field | Correlation Coefficient | −0.130 | −0.174 | −0.099 | −0.027 | −0.385 ** | −0.021 | 1.000 | −0.049 |
| | Sig. (2-tailed) | 0.320 | 0.183 | 0.453 | 0.836 | 0.002 | 0.872 | | 0.712 |
| | N | 60 | 60 | 60 | 60 | 60 | 60 | 60 | 60 |
| The monthly energy consumption | Correlation Coefficient | 0.176 | −0.214 | 0.151 | 0.021 | 0.263 * | −0.002 | −0.049 | 1.000 |
| | Sig. (2-tailed) | 0.180 | 0.100 | 0.249 | 0.876 | 0.043 | 0.987 | 0.712 | |
| | N | 60 | 60 | 60 | 60 | 60 | 60 | 60 | 60 |

Source: authors' elaboration on sample data. ** Correlation is significant at the 0.01 level (2-tailed). * Correlation is significant at the 0.05 level (2-tailed).

Nevertheless, regarding the first hypothesis, it is to be noticed that there is a low correlation between experience and level of knowledge (0.305), and also between the level of knowledge and attitude (0.348), which proves that the first part of the hypothesis is false. But, taking into consideration that more than half (56%) of the respondents are willing to sign PPAs, can assume that even if the levels of knowledge nor experience are at low levels, sellers can see the benefits of signing PPAs. To check the second part of the hypothesis, multiple linear regression was performed using IBM SPSS Statistics for Windows (Version 29.0.0.0, Software vendor: https://www.ibm.com/products/spss-statistics, accessed on 7 March 2024). The results are presented below, in Tables 6 and 7:

**Table 6.** ANOVA Table with Attitude regression for seller.

| | | | | | | |
|---|---|---|---|---|---|---|
| | | **ANOVA** [a] | | | | |
| **Model** | | **Sum of Squares** | **df** | **Mean Square** | **F** | **Sig.** |
| 1 | Regression | 6.814 | 2 | 3.407 | 4.100 | 0.022 [b] |
| | Residual | 47.369 | 57 | 0.831 | | |
| | Total | 54.183 | 59 | | | |
| 2 | Regression | 6.803 | 1 | 6.803 | 8.328 | 0.005 [c] |
| | Residual | 47.380 | 58 | 0.817 | | |
| | Total | 54.183 | 59 | | | |

[a] Dependent variable: attitude; [b] predictors: (constant), experience, level of knowledge; [c] predictors: (constant), level of knowledge.

**Table 7.** Coefficients Table with Attitude regression for seller.

| | | Coefficients [a] | | | | | | | |
| | Model | Unstandardised Coefficients | | Standardised Coefficients | t | Sig. | 95.0% Confidence Interval for B | |
| | | B | Std. Error | Beta | | | Lower Bound | Upper Bound |
|---|---|---|---|---|---|---|---|---|
| 1 | (Constant) | 2.582 | 0.316 | | 8.177 | <0.001 | 1.950 | 3.214 |
| | Level of knowledge | 0.321 | 0.118 | 0.359 | 2.732 | 0.008 | 0.086 | 0.557 |
| | Experience | −0.012 | 0.106 | −0.015 | −0.113 | 0.911 | −0.223 | 0.199 |
| 2 | (Constant) | 2.565 | 0.275 | | 9.330 | <0.001 | 2.015 | 3.115 |
| | Level of knowledge | 0.317 | 0.110 | 0.354 | 2.886 | 0.005 | 0.097 | 0.537 |

[a] Dependent variable: attitude. Source: authors' elaboration on sample data.

By analysing the results of regression, even if Sig. has a good value (under 0.05), we can see that this regression is applicable to a small number of respondents, with the residuals being 47.38 out of 54.183 of the sum of squares.

Regarding the second hypothesis, as we have showed before, it can be considered false, both from the buyer's and seller's point of view, by analysing correlations between the development of the PPA market and the involvement of the state, where the coefficient is 0.094 for buyers and 0.230 for sellers. From our point of view, this can be interpreted in two ways:

There are other factors, from the private sector, that affect the PPA market, which can be easily explained by the fact that when we are talking about energy from renewable resources, most energy is produced by the private sector (except Hydro).

The PPA market in Romania is very lowly developed (proved also by Figure 4), so the respondents cannot find the best way to increase development, putting this situation partially on the state's shoulders, asking for more support from its side (67% for seller and 51% for buyer).

The third hypothesis will have a common analysis for both the buyer and the seller and to prove this, using IBM SPSS Statistics for Windows (Version 29.0.0.0, Software vendor: https://www.ibm.com/products/spss-statistics, accessed on 7 March 2024), a simple linear regression was performed, taking experience as the dependent variable and parties as the independent variable. The result is presented below in Tables 8–10:

**Table 8.** Model Summary for Experience regression.

| | | | | Model Summary | | | | | |
| Model | R | R Square | Adjusted R Square | Std. Error of the Estimate | R Square Change | Change Statistics | | | |
| | | | | | | F Change | df1 | df2 | Sig. F Change |
|---|---|---|---|---|---|---|---|---|---|
| 1 | 0.137 [a] | 0.019 | 0.015 | 1.168 | 0.019 | 4.425 | 1 | 230 | 0.037 |

[a] Predictors: (constant), parties.

Looking at the model Summary of this regression and also the correlations matrix performed using parametric (Pearson) and nonparametric (Kendall's tau and Spearman's rho) bivariate correlations, it is obvious that there is no connection between experience and party, so we can say that hypothesis 3 is confirmed. Sadly, one of the conclusions of this analysis can be that both parties present in the Romanian PPA market have little interest in PPAs, so that experience is based on individuals, not companies/parties.

Although the last hypothesis is not confirmed, with there being no correlation between monthly consumption and attitude, where the correlation coefficient is 0.214, taking into consideration that the respondents are in favour of PPAs (more than 55% in both cases), we can conclude that PPAs are beneficial for all parties, independent of monthly consumption.

**Table 9.** Pearson Correlations between Parties and Experience.

| | Correlations | Parties | Experience |
|---|---|---|---|
| | | **Parties** | **Experience** |
| Parties | Pearson Correlation | 1 | 0.137 * |
| | Sig. (2-tailed) | | 0.037 |
| | N | 232 | 232 |
| Experience | Pearson Correlation | 0.137 * | 1 |
| | Sig. (2-tailed) | 0.037 | |
| | N | 232 | 232 |

*. Correlation is significant at the 0.05 level (2-tailed).

**Table 10.** Kendall's tau and Spearman's rho Correlations between Parties and Experience.

| | | Correlations | Parties | Experience |
|---|---|---|---|---|
| | | | **Parties** | **Experience** |
| Kendall's tau | Parties | Correlation Coefficient | 1.000 | 0.119 * |
| | | Sig. (2-tailed) | | 0.047 |
| | | N | 232 | 232 |
| | Experience | Correlation Coefficient | 0.119 * | 1.000 |
| | | Sig. (2-tailed) | 0.047 | |
| | | N | 232 | 232 |
| Spearman's rho | Parties | Correlation Coefficient | 1.000 | 0.131 * |
| | | Sig. (2-tailed) | | 0.047 |
| | | N | 232 | 232 |
| | Experience | Correlation Coefficient | 0.131 * | 1.000 |
| | | Sig. (2-tailed) | 0.047 | |
| | | N | 232 | 232 |

* Correlation is significant at the 0.05 level (2-tailed). Source: authors' elaboration on sample data.

## 5. Conclusions

This study has investigated the main factors which either help or prevent the promotion and development of PPAs on the road to achieving climate neutrality and meeting the related sustainability goals. Entering into contractual relations within a PPA depends very much on the measures that will be taken, especially at the national level, for the development and implementation of PPAs.

As we mentioned above, a power purchase agreement (PPA) contracting mechanism is a tool that provides long-term predictability for consumers, as well as energy producers and suppliers.

The accelerated growth in renewable energy investment will continue through the commissioning of a new capacity, and PPAs will play a protection role in this regard for energy consumers and suppliers from future price increases, potential market manipulation, and will help make the industry cleaner and more competitive.

The research results indicate that at the moment, there is still no clear understanding of the PPA mechanism as an instrument for achieving climate neutrality and meeting the related sustainability goals. The Spearman correlations showed that there are no correlations between the level of knowledge, attitude, and monthly energy consumption, all analysed from the buyer's and from the seller's point of view. It must be highlighted that even though there may be some state involvement, this is not enough, with more actions being needed.

Also, although the results showed that the level of knowledge is quite low, the attitude is favourable, which shows that there is an intention to enter into such contractual relationships that offer consumers access to competitive prices for renewable and non-fossil energy electricity.

However, despite the intention that exists to enter into contractual relations within a PPA, there is no legal framework that allows for the start of these relations, there are no legal measures to facilitate this, the procedure is a heavy and unattractive one, many respondents are afraid to sign a PPA, because they do not master this mechanism very well, and they do not know what to expect from such a contract; therefore, they refuse to sign a PPA. According to Stanitsas [38], utilities have had to develop complex procedures to adapt to an increasing proportion of fluctuating renewable energy costs. On the other hand, corporates are more willing to sign long-term PPAs as renewables become the main energy technology for large energy corporations and as new renewable energy suppliers enter the market. It is essential that they are aware of the risks they may be exposing themselves to. However, it should not be forgotten that regardless of the industry in which the off-taker operates, the creation/development of an appropriate project from the point of view of sustainability can be considered, which means initial planning, which must first be analysed in terms of environmental impact and the resources used. This is the moment when determining both the sources of energy used and how to reduce electricity consumption is a priority.

But in Romania, to build a sustainable PPA market, a fast development of economical/political environments is needed in order to create a solid basis for long-term PPA development, as well as increasing clean energy production. As seen before, this would be not enough without social development, which involves implementing training courses, conferences, and programs to show and explain the benefits that PPA can produce in terms of clean energy production, such as decarbonisation, attracting investments in clean energy, and lower prices.

A criterion to demonstrate the sustainability of PPAs is long-term planning, which plays a major role in increasing the level of knowledge and the positive attitude of people towards involving themselves in signing a PPA. By promoting PPAs and explaining/proving their advantages, not only would society's level of knowledge increase, which will obviously improve the attitude towards entering into such contractual relationships, but the level of the PPA market would also increase significantly, an aspect that will highlight a demand, and therefore an increasingly rapid development of the electricity market from renewable energy sources. The impact on the objectives imposed by the European Union, implicitly desired by the Member States, will be obvious and will be evidenced by clear economic and financial stability, marked by fixed and determined prices, which will become the main benefit of concluding a PPA.

Therefore, if this base is established in the short term, it will be much easier to create a sustainable PPA market through planning, involving more companies, analysing results, and reacting quickly to any negative impacts and setting new and higher decarbonisation targets.

**Author Contributions:** Conceptualization, A.T. and E.N.; methodology, A.T., E.N., C.S. and C.A.; software, L.F. and E.N.; validation, A.T., C.S. and C.A.; formal analysis, C.A.; investigation, E.N.; resources, C.S. and L.F.; data curation, A.T. and E.N.; writing—original draft preparation, A.T., E.N. and C.S.; writing—review and editing, A.T., E.N. and C.A.; visualization, C.S. and L.F.; supervision, A.T.; project administration, E.N. All authors have read and agreed to the published version of the manuscript.

**Funding:** This research received no external funding.

**Institutional Review Board Statement:** The study did not require ethical approval.

**Informed Consent Statement:** Informed consent was obtained from all subjects involved in the study.

**Data Availability Statement:** Data are contained within the article.

**Conflicts of Interest:** The authors declare no conflicts of interest.

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
