# Peer review of "Identification and Analysis of the Key Factors That Influence Power Purchase Agreements on the Road to Sustainable Energy Development"

_sustainability, doi:10.3390/su16083202_

Round 1

Reviewer 1 Report

Comments and Suggestions for Authors

Thank you for submitting you article to MDPI. Please address the following comments to improve the article.

1. Expand the literature review ? mention the previous work in the area and clearly state the paper’s contribution

2. Line 169, correct it to ‘he can’.

3. In the description for the details in Table 1, can you elaborate the terms ‘quality of the buyer’ and ‘quality of the seller’, ‘frequency’ ,‘percent’ , ‘signing before PPA’

4. Line 314, Correct  it to ‘Percentage’

5. Define the acronym Tf

6. Explain how Spearman’s rho correlation is calculated

7. Explain Sig. (2-tailed) 

8. What is N in Table 5?

Comments on the Quality of English Language

Minor corrections are required. 

Author Response

Thank you for your remarks. In the revised paper we tried to respond with a green color to all your comments.

We expand the literature review and clarify the paper’s contribution.

We explained the terms used for table 1, the Spearman rho correlation  and made also other corrections.

Reviewer 2 Report

Comments and Suggestions for Authors

I have read a very interesting article prepared by you entitled Identification and analysis of the key factors which prevent or contribute to the development of Power Purchase Agreements on the road to climate neutrality and meeting the related sustainability goals.

It treats very interesting issues. In order to improve the quality of the article, it recommends:

1. Structure 

A. The introduction section should include the hypotheses, and the purpose and specific objectives of the study should be stated.

B. Please improve the visual form of the text as the current one lacks paragraphs and content division. Also, the footnotes are not drawn up according to MDPI rules.

2 Literature review

I recommend the literature review as weak and it needs to be restructured. Please embed the study in economic theory. Some sections lack reference to ligterature, i.e. there is a sub-text but no footnotes. Please rethink this review and draft according to the requirements of the journal. 

3. It may be displaying badly at my place, but for the final version please bring the tables and images in line with MDPI requirements.

General information about the general principles contained in this article.

Author Response

Thank you for your feedback. In the revised paper we tried to respond with a green color to all your comments.

We included the hypotheses, and the purpose and the specific objectives of the study in the introduction part.

We improved the literature review and its reference structure according to MDPI requirements and made connections with the economic theory. 

Reviewer 3 Report

Comments and Suggestions for Authors

This article “Identification and analysis of the key factors which prevent or contribute to the development of Power Purchase Agreements on the road to climate neutrality and meeting the related sustainability goals”, it aims to identify and analyze the main factors that help or prevent the promotion and development of a PPA, to meet the sustainability goals by promoting a green economy. 

While the overall organization is commendable, there are minor areas for improvement:

1. The title's length could be streamlined for clarity and conciseness.

2. The research motivation should be expounded upon to underscore the significance of the employed technologies and tools.

3. Strengthening the application background is crucial for enhancing the paper's relevance to practical scenarios.

4. Expanding the conclusion section and addressing any limitations encountered would provide a comprehensive overview.

5. It's advisable to meticulously review the language and expressions to enhance reader comprehension.

6. Figure 5 is not clear

7. Page 8, this tabe 1 is better to present it as a figure, same to Table 4.

8. The introduction has to present how transport plays an important role in providing and achieving sustainability goals, recent following works are recommended: -- A novel Parsimonious Spherical Fuzzy Analytic Hierarchy Process for Sustainable Urban Transport Solutions, Engineering Applications of Artificial Intelligence.--- A Novel Parsimonious Best Worst Method for Evaluating Travel Mode Choice. IEEE Access.

Author Response

Thank you for these feedback. We understand your concerns. In the revised paper, we tried to emphasize better all mentioned comments. We used a green color for all improvements.

Reviewer 4 Report

Comments and Suggestions for Authors

I suggest to implement and upgrade the "Introduction" paragrapher by more recent contribution. Eg. A. quote the Paris Conference, but no the Glasgow and Doa ones. Also the  base literature moving the initial part of paper can be implemented.

The topic, Power Purchase Agreements, is illustrated on the base of related literature. I have suggested to implement it quoting important conference to have more connection with the Sustainable Development Goals. Non more than this, because the paper is focused on PPA. paragraphers 3 and 4 are more interesting parts. Probably, the paper's title could be revisited a bit. Applied method is correct
Apply to the Romania test case and results seem correct. They use national data. Introduction can be revisited and updaded. Paragrapher 1 is more useful to understand the general framework. Spearman’s rho correlation matrix is well known and correctly applied.

Author Response

Thank you for for the feedback.  We understand your concerns in this regard.  Starting with the updated introduction we made with a green color al improvements.

Round 2

Reviewer 3 Report

Comments and Suggestions for Authors

In this revised version, all my concerns have been addressed well by the authors. I recommend accepting this version for publishing in this journal.